# Balance Differences between North and South American Older Adults: A Cross-Sectional, Age and Sex Matched Study

**DOI:** 10.3390/healthcare10030499

**Published:** 2022-03-09

**Authors:** Matheus Almeida Souza, Daniel Goble, Paige Arney, Edgar Ramos Vieira, Gabriela Silveira-Nunes, Leonardo Intelangelo, Michelle Almeida Barbosa, Alexandre Carvalho Barbosa

**Affiliations:** 1Musculoskeletal Research Group—NIME, Department of Physical Therapy, Federal University of Juiz de Fora, Juiz de Fora 36036-900, Brazil; malmeida_1812@hotmail.com (M.A.S.); michelle.almeida@visitante.ufjf.br (M.A.B.); 2Department of Exercise Science, Oakland University, Rochester, MI 48309, USA; dgoble@oakland.edu (D.G.); pmarney@oakland.edu (P.A.); 3Department of Physical Therapy, Florida International University, Miami, FL 33179, USA; evieira@fiu.edu; 4Musculoskeletal Research Group—NIME, Department of Medicine, Federal University of Juiz de Fora, Governador Valadares 35010-180, Brazil; gabriela.abreu@ufjf.edu.br; 5Musculoskeletal Research Unit—UIM, Department of Physical Therapy, University Center for Assistance, Teaching and Research—CUADI, Universidad del Gran Rosario—UGR, Corrientes 1254, Argentina; leonardo.intelangelo@gmail.com

**Keywords:** aging, accidental falls, frail elderly, postural balance

## Abstract

This study aimed to characterize the risk of falling in low-, moderate- and high-risk participants from two different geographical locations using a portable force-plate. A sample of 390 older adults from South and North America were matched for age, sex, height and weight. All participants performed a standardized balance assessment using a force plate. Participants were classified in low, moderate and high risk of falling. No differences were observed between South and North American men, nor comparing North American men and women. South American women showed the significantly shorter center of pressure path length compared to other groups. The majority of the sample was categorized as having low risk of falling (male: 65.69% and female: 61.87%), with no differences between men and women. Moreover, no differences were found between North vs. South Americans, nor between male and female groups compared separately. In conclusion, South American women had better balance compatible with the status of the 50–59 years’ normative age-range. The prevalence of low falls risk was~61–65%; the prevalence of moderate to high risk was~16–19%. The frequency of fall risk did not differ significantly between North and South Americans, nor between males and females.

## 1. Introduction

The worldwide expansion of the older adult population is a well-established trend [1,2]. Aging is associated with declines in neuromuscular function and physical performance [3,4], increased risk of falling, hospitalization, fractures and mortality [3,5,6]. The annual prevalence of falls among older adults in the two largest South American countries has been estimated to be 27% (Brazil) and 28.5% (Argentina), respectively. This is similar to the rate seen in the United States (28.7%) [7,8]. The personal and economic burden of falling is significant, representing a total cost of billions of dollars to the public healthcare systems [9,10]. Examining risk factors for falls, such as impaired balance, may help identify those at risk and alleviate the physical and psychological impacts of falls. 

Balance depends on the ability to maintain the center of mass over one’s base of support and is achieved via coordination of complex sensory, motor, and biomechanical processes [11,12]. For older adults, balance is especially important in order to recover from disturbances and to prevent falls [13,14,15]. Poor balance is associated with aging-related physical decline [16,17], with impaired balance variables being correlated to increased falls risk [18,19]. Poor balance and impaired mobility are associated with a 33% increased risk of recurrent falls in older adults [20]. 

Balance can be accurately and reliably measured using a force plate device [21,22]. Force plates measure postural sway based on center of pressure (COP) oscillations, which is equal to the weighted average of forces created when people stand on the force plate for a previously set amount of time [23,24]. Older adults who fall repeatedly have a greater magnitude of COP oscillations than those without a history of falling [25,26]. Furthermore, older people with increased postural sway present a greater likelihood of falling [21,25,26]. 

Force plate-based balance assessments in older adults have been somewhat limited due to cost ($10,000–$100,000), technical difficulties, and time/training to process and extract data from the raw signals [27,28]. Force plates often lack user-friendly software that are time efficient, reliable, and intuitive [29]. However, recent developments led to reliable and low-cost force plates with user-friendly software [30]. The Balance Tracking System (BTrackS, Balance Tracking System, San Diego, CA, USA) is a low-cost (<$2000) and portable (<7 kg, USB-powered) force plate (FDA class 1 registered device) with user-friendly software for balance assessment that has been shown to be valid and reliable for testing community-dwelling older adults [31]. BTrackS has also been shown to accurately detect the risk of falling in older adults based on their balance indicators [29].

The largest known normative database of older adult balance was created using the BTrackS balance and fall risk test protocol [32]. This protocol measures the postural sway of an individual while standing still with eyes closed for three trials of 20-s. The numerical results are converted in a categorical model according to a predefined range of body sway, allowing early detection of balance impairments [29,31]. According to a manufacturer report (https://balancetrackingsystems.com/wp-content/uploads/2019/05/Validating-BTrackS-FRA.pdf (accessed on 20 November 2021)), the low-risk category equates to a 29% likelihood of falling in the next 12 months. This likelihood rises to 42% for those in the moderate category, and to 51% for those classified in the high risk category [29].

One limitation of the BTrackS system is that the normative data utilized is almost exclusively based on values from the United States population. Considering the lack of studies comparing balance data in different locations, the primary aim of the present study was to compare the balance of a representative sample of healthy North and South American older adults by sex. Additionally, this study aimed to characterize the risk of falling in low-, moderate- and high-risk categories using BTrackS criteria. It was hypothesized that balance and falls risk would not be significantly different between North and South America older adults, nor between older men and women.

## 2. Materials and Methods

### 2.1. Participants

The present study was a cross-sectional from a secondary analysis of the balance data collected from previous studies with older adults using the same equipment and procedure for all participants. A total of 390 older adults (≥60 years of age) of both sexes were included (Table 1). Participants were taken from two geographical locations (North vs. South America) and closely matched for age, sex, height and weight. The sample selection also considered the previously reported higher frequency of falls in women compared to their male counterpart [7,33,34]. Specifically, the participants were selected from three of the largest countries on the South and North American continents: Brazil and Argentina (South America), as well as the United States (North America). Data for the North and South American sample was sub-sampled from an existing database of balance performance in healthy, community dwelling individuals. Data was collected at multiple sites, including community centers, fitness facilities and wellness/health fairs. The participants were selected in a deliberate fashion so as to match the age, height and weight characteristics of each individual in the South American sample. Four comparison groups were, therefore, determined from the sample including: South American Men, North American Men, South American Women, and North American Women. 

All procedures were performed on-site without any prior practice. Attendance was taken as compliance with the assessment protocol. No co-interventions were performed in either group; no adverse effects were reported by any participant during any procedure. Prior to testing, the participants were familiarized with all physical assessment procedures. The sample was physically independent (level 3 or 4 on Functional Status) [35], and cognitively able to understand the procedures (score > 21 on the Mini-Mental State Examination for people with low education) [36]. Exclusion criteria were incidence of cardiovascular disease, unstable proliferative retinopathy, end-stage renal disease, and uncontrolled hypertension [37]. All participants were free of any knee or hip injury that could affect their balance. The participants were also cleared from medication that could cause dizziness as a side effect. 

Based on a post hoc power calculation an obtained effect size (ε^2^) of 0.252, with an alpha level of 5%, was determined for the total sample size of 390; an actual power of 0.992 was determined for the four individual groups. The sample power was calculated using G-POWER™ software (Version 3.1.5, Franz Faul, Universitat Kiel, Kiel, Germany). Procedures for this study were approved by the institutional review boards from all countries involved (Argentinian protocol number 38/16 UGR-Rosario; Brazilian protocol number 59862316.7.0000.5147; USA protocol number 2203100).

### 2.2. Equipment and Procedure

The BTrackS Balance Plate (Balance Tracking System, San Diego, CA, USA) was used to assess balance. This device consists of a force platform (40 × 60 cm, sampling frequency of 25 Hz) with four implanted strain gauges that determine the center-of-pression (COP) excursion distance when stood upon. The BTrackS sampling frequency satisfied the Nyquist theorem for the slow (<10 Hz) COP changes measured in the present study. Previous work has shown BTrackS to perform with the same accuracy/precision as a laboratory-grade force platform [30]. Prior to testing, the force platform was leveled via adjustable legs, and connected to the computer through a USB cable, which also provided power to the plate. 

The procedure for testing required all participants to perform four, 20 s two-legged stance trials (Figure 1). The first trial was for familiarization and was discarded before analysis. The remaining three, non-familiarization trials were used to determine the result. Trials were conducted in a closed room to reduce noise and disturbances. Participants were instructed to firstly look straight ahead and to stand as still as possible on the BTrackS Balance Plate with eyes closed, hands on hips and feet shoulder width apart. Participants were monitored by the tester during all trials to avoid a fall episode.

The result for each test was calculated by the BTrackS Assess Balance software, equivalent to the average total COP path length in centimeters (cm) across accountable trials. COP path length is a proxy for postural sway magnitude where larger BBT values are indicative of greater postural sway [21,32,38]. Path length was determined by first quantifying the distance between successive registered COP locations according to the following formula:distance = ([COP_x2_ − COP_x1_]^2^ + [COP_y2_ − COP_y1_]^2^)^0.5^
where COP_x2_ and COP_x1_ are adjacent time-points in the COP_x_ (medial/lateral) time series, and COP_y2_ and COP_y1_ are adjacent time-points in the COP_y_ (anterior/posterior) time series. The sum of all distances was then added together to obtain the total path length. Participants were also classified into one of three fall risk categories: low risk (postural sway range: 0–32 cm for men and 0–30 cm for women); moderate risk (postural sway range: 33–40 cm for men and 31–38 cm for women); and high risk (postural sway range: 41+ for men and 39+ for women) [27].

### 2.3. Statistical Analysis

Shapiro–Wilk and the Levene’s tests were first performed to assess the normality and equality of variances for each group tested. Given that normality was not achieved, descriptives for this study are presented as median, minimum and maximum values. For statistical comparisons of group data, non-parametric Kruskall–Wallis tests were used with Dwass–Steel–Critchlow–Fligner post hoc correction to protect against multiple comparisons. The between-group fall risk categorization frequencies (low-, moderate- and high-risk) were evaluated using a chi-square test for ordinal comparisons. Significance was set at the *p* < 0.05 level for all tests. Standardized differences for comparisons were analyzed using the rank biserial correlation effect size (ES). The magnitude of the ES was qualitatively interpreted using the following thresholds: 0.01–0.19; small: 0.20–0.49; moderate: 0.50–0.79; large: 0.8–1.19; very large: 1.2–1.99; and huge: >2 [39]. All analysis was performed using the JAMOVI software (version 1.6.15.0, 2021; retrieved from https://www.jamovi.org (accessed on 4 September 2021)).

## 3. Results

No differences were observed between men (South American men: median = 32 cm (minimum–maximum: 16–75 cm), Northern American men: 33 cm (16–79 cm); *p* = 0.99; ES = 0.001), nor comparing North American men and women (North American women: 27 cm (10–115 cm); *p* = 0.14; ES = 0.21). However, South American women showed the lowest path length compared to other groups (South American women: 24 cm (10–70 cm); all *p* < 0.05), with effect sizes ranging from small to moderate (Figure 2). 

Considering the overall sample, the majority of the sample was categorized as low risk of falling (male: 65.69% and female: 61.87%). The frequency of the low-risk category was higher (χ^2^ = 8.32, *p* = 0.016) compared to the moderate- (men: 16.34% and women: 18.18%) and high-risk categories (male: 17.97% and female: 19.95%). No differences were observed comparing men vs. women. Moreover, no differences were found between North vs. South America (Table 2), nor for categories’ frequencies when male or female groups were compared separately. 

## 4. Discussion

The primary focus of the present study was to compare the balance of North vs. South American older adults grouped by sex using the same device (i.e., BTrackS). Only South American women showed differences compared to other groups, with improved postural control (i.e., better balance) at a small-to-moderate effect size. There were no differences seen in the probability of fall risk across sex and geographic region. To the best of our knowledge, this is the first study comparing the balance of a representative sample of older adults from distinct continents using a standardized protocol with the same device. These results suggest that balance ability may differ to some extent across large geographical areas. In this case, pre-existing balance norms would require some adjustment based on geographic location and sex demographics.

In contrast to our results, one previous study showed decreased balance control in older women compared to men [40]. The authors also reported that older women had greater COP displacement but less sway velocity than their male counterpart, especially in mediolateral direction. However, the protocol used to assess the balance included only 2 trials with open eyes, 30 s of data collection and 5 min of rest between trials. The present protocol was performed with closed eyes, as vision is a major contributor to postural stability [41]. The visual system inputs information on head position relative to the surrounding environment, which might be used in a feedforward fashion to anticipate a loss in balance [42]. On the other hand, the closed eyes task imposes greater efforts on the proprioceptive system, becoming the main source of neural feedback during an imbalance situation [13,42,43]. Further, the current protocol was extensively validated in thousands of subjects, with consistent results showing greater postural sway in older men compared to age-matched older women [21,30,42,44]. One of these previous studies compared the balance results of 812 older men compared to 2341 older women, with a mean difference of ~ 8 cm on the path length variable [21]. The difference was consistent across all three ranges of older age (60–70, 70–80, and 80+ years).

The male groups’ overall results, along with the North American women, are in accordance to those expected for their age [21]. The South American women, however, showed a body sway compatible with younger normative data (~50–59 years). Due to the study design, no cause–effect can be inferred from the present results. Nevertheless, it is well known that some factors may influence the balance outcomes in older adults. According to an updated meta-analysis with 19,478 participants, the greater effects were seen from exercise protocols that challenged balance and that involved more than 3 h/week of exercise [45,46,47]. Thus, a possible explanation for the present results could be the South American women’s engagement in physical activities impacting the musculoskeletal development and the consequent balance improvements. However, the Brazilian Longitudinal Study of Aging (ELSI-Brazil), a nationally representative, population-based cohort study of persons aged 50 years or more, accounted 8736 individuals (53.5% were women) in a study that showed that declines in physical activity levels with increasing age were significantly more evident among women [48]. Another possible explanation for the present findings is the possible higher rate of falls and frailty in North American older adults (15%) compared to South American older adults (9%) [49,50]. Physical frailty is characterized by weakness, slowness, and reduced muscle mass, yet with preserved ability to move independently [51]. In this sense, the balance might be impaired due to inability of the sensory system combined with muscle inefficiency to recover from perturbations on the COP placed by external forces ( e.g., gravity, body mass distribution, inertia) acting over stabilizing internal torques (mainly muscle and ligament forces). [52,53] As an example, decreased strength is related to diminished postural control, which is associated with increased risk of falls [53]. Impaired balance is also predictive of bad outcomes in older adults, such as frailty and higher prevalence of falls [14,54,55]. Considering the overall prevalence of falls, South American older adults have an average rate of 27% while North American older adults have a higher rate of 28.7% (33.9% in certain states) [8,56,57].

A secondary aim of the present study was to examine the frequency of low, moderate and high risk of falling. Despite the higher percentage of overall older people under low risk of falling independent of location, an important proportion of participants was classified as moderate or high risk of falling (~35%). Balance is maintained or restored in the hip strategy mainly by movement of the body around the hip and trunk, while, in the ankle strategy, it is restored mainly by the movement of the body around the ankle as a single inverted pendulum with minimal movement about superior joints [58,59]. In an unexpected situation of imbalance, an older adult at high risk of falling tends to use the hip strategy to maintain postural stability compared to others under low risk of falling [60]. The use of such a strategy may increase the confidence, as the COP tends to move faster to stay within the base of support [60]. Mechanically, the foot’s ability to exert torque in contact with the support surface limits the ankle strategy, as distal muscles are acting far from the body’s COP to stabilize the full body sway [59]. Although both strategies are biomechanically different, they have the main goal to maintain the forward progression and the upright balance.

As individuals in the moderate and high risk category are recommended to be targeted for balance intervention programs [29], health care professionals may benefit their older patients with early diagnostics and active intervention to prevent bad outcomes that follows an episode of fall (e.g., head trauma, spine and hip fractures, hospitalization) [61,62]. Moreover, a comprehensive balance screening focused on the lower limb and trunk’s muscles is essential to target the segments in need of care, as the current protocol is performed using the ankle strategy. Thus, the classification is also based on the level of COP displacement considering only this strategy.

An important limitation of the present study is the cross-sectional design, which does not allow cause–effect inferences. Further investigation is needed to determine changes over time in individuals receiving interventions to improve the balance status. It is also important to address that the present study only assessed the postural balance, one aspect of the functionality. The level of physical function of the sample was also a limitation. Although the falling status was properly categorized between groups, all participants were at a high functioning level. Institutionalized older women or those presenting high levels of frailty may show distinct balance patterns.

## 5. Conclusions

South and North American men have the same balance status, as well as the North American women. On the other hand, South American women showed improved balance compatible with the status of the previous normative age-range (50–59 years). A higher percentage of overall older people under low risk of falling was observed independent of location, but an important proportion of participants was classified as moderate or high risk of falling. The balance categories frequencies did not vary considering the location (South vs. North), nor considering the gender factor.

## Figures and Tables

**Figure 1 healthcare-10-00499-f001:**
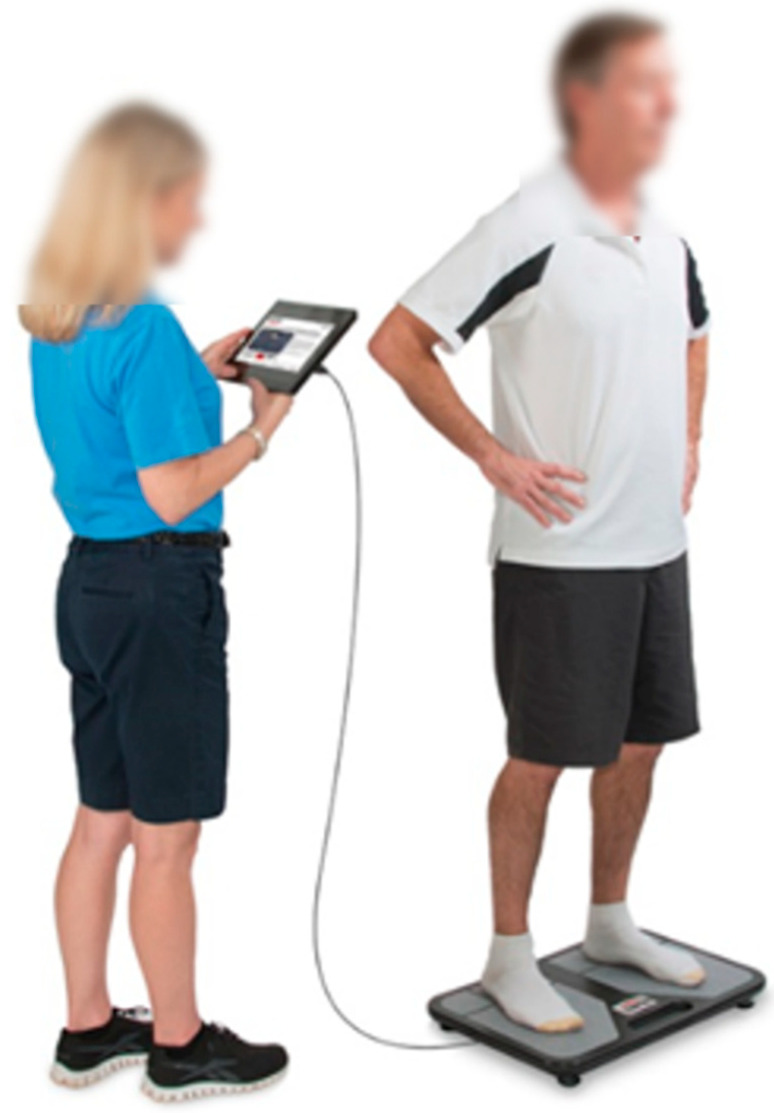
BTracks testing setup for a fall risk assessment.

**Figure 2 healthcare-10-00499-f002:**
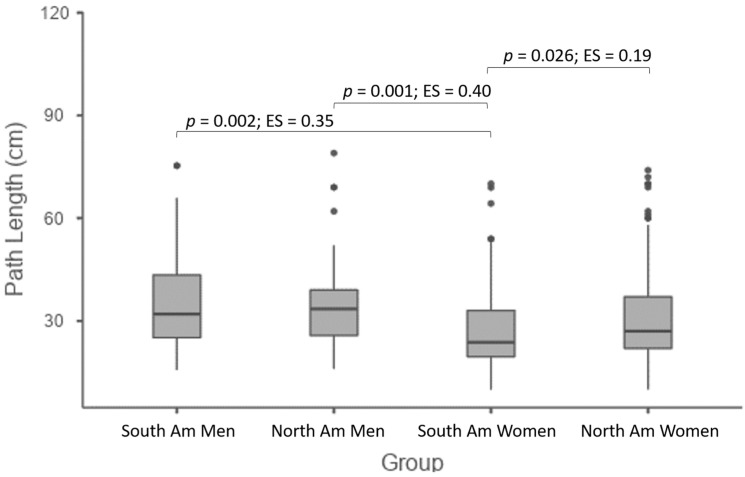
Box plot of pairwise comparisons. Box trace = median values; box limits = minimum–maximum; vertical trace = standard deviation; dots = outliers; ES = effect size. Significant differences assigned.

**Table 1 healthcare-10-00499-t001:** Participants’ characteristics in median (minimum–maximum). Mann-Whitney non-parametric test results.

Characteristics	Men			Women		
	South	North	*p*	South	North	*p*
*n*	44	44	-	151	151	-
Age (years)	70 (61–86)	71 (61–87)	0.89	70 (60–98)	71 (60–99)	0.95
Height (cm)	167 (105–184)	168 (155–185)	0.87	155 (137–176)	155 (124–175)	0.56
Weight (kg)	74.5 (52–94)	76.5 (50–98)	0.59	64 (42–98)	64 (43–127)	0.94

**Table 2 healthcare-10-00499-t002:** Risk of falling frequencies in% (and absolute) of the sample per gender.

Gender	Group	Risk of Falling			Difference
		Low	Moderate	High	
Male	South	27% (23)	6.67% (6)	16.67% (15)	χ^2^ = 4.40, *p* = 0.111
	North	23% (21)	15.56% (13)	11.11% (10)	
Female	South	34.64% (104)	8.50% (26)	6.86% (21)	χ^2^ = 3.81, *p* = 0.149
	North	30.72% (92)	8.17% (25)	11.11% (34)	

## Data Availability

Raw data is available from the following doi: 10.17632/cyg8vcxytn.1.

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
