# Peer review of "Balance Differences between North and South American Older Adults: A Cross-Sectional, Age and Sex Matched Study"

_healthcare, 2022, doi:10.3390/healthcare10030499_

Round 1

Reviewer 1 Report

Thank you for the revisions you have made. They improve the paper's clarity.
Best wishes for your publication.

Author Response

Thank you for your time and effort to improve our manuscript's clarity.

Regards.

Reviewer 2 Report

In this research, the authors investigated balance differences between North and South American older adults. The results showed no significant differences between South and North American older adults. The number of samples is 390 people, which indicates the value of this research. However, the following problems, especially in originality, need to be revised.

Main problems:

  1. The authors provide a hypothesis that balance and fall risk would not be significantly different between north and south American older adults, nor between older men and women (Line 83). Why do authors give and try to verify this hypothesis? Are there any references that lead to the hypothesis? Please give more details about this issue because this is the research purpose.

  1. One interesting result is that South American women had better balance compatible with the status of the 50-59 years’ normative age range, demonstrating the originality of this research. There are some discussions in Line 220 to explain the newfound. The authors state a possibility that South American women’s engagement in physical activities impacts musculoskeletal development and the consequent balance improvements. However, it cannot explain that only South American women are different from others. Please give other references such as South American women spend more time in physical activities than others to support the guess.

  1. The reviewer understands the statistical results provide meaningful conclusions. However, the mechanism behind the statistical results also needs to be stressed more deeply to improve the value of this study. For example, is there any reference support or point out that reason may be a factor to influence unbalanced movements biomedically or biomechanically? If the authors discuss from these points of view will be highly preferable.

One minor problem:

Reference numbers need to be moved to in front of “,” or “.”.

Author Response

In this research, the authors investigated balance differences between North and South American older adults. The results showed no significant differences between South and North American older adults. The number of samples is 390 people, which indicates the value of this research. However, the following problems, especially in originality, need to be revised.

Main problems:

The authors provide a hypothesis that balance and fall risk would not be significantly different between north and south American older adults, nor between older men and women (Line 83). Why do authors give and try to verify this hypothesis? Are there any references that lead to the hypothesis? Please give more details about this issue because this is the research purpose.

ANSWER: Thank you for your comment and attention. To confirm the null hypothesis was always the primary goal of our study, and we pursued this path until the statistics showed another outcome. As no previous study have compared those populations, the novelty of our study was to establish such differences to provide more information regarding their balance status so clinicians may account them on their prevention and intervention programs.

One interesting result is that South American women had better balance compatible with the status of the 50-59 years’ normative age range, demonstrating the originality of this research. There are some discussions in Line 220 to explain the newfound. The authors state a possibility that South American women’s engagement in physical activities impacts musculoskeletal development and the consequent balance improvements. However, it cannot explain that only South American women are different from others. Please give other references such as South American women spend more time in physical activities than others to support the guess.

ANSWER: Thank you very much for your comment. After searching for references we have discovered that this cannot be an explanation, as a large sampled study showed the opposite. We added the reference and the explanation in our current version. Thus, the second hypothesis (with references!) that North American has higher rates of falling and frailty would be more plausible for our results. Thanks again for your attention.

The reviewer understands the statistical results provide meaningful conclusions. However, the mechanism behind the statistical results also needs to be stressed more deeply to improve the value of this study. For example, is there any reference support or point out that reason may be a factor to influence unbalanced movements biomedically or biomechanically? If the authors discuss from these points of view will be highly preferable.

ANSWER: Thank you for your comment. We already have inserted some biomechanical topics in our discussion. But as we mentioned in our limitation section, this is a cross-sectional study design. Thus, we are very limited to our data. Any inference or cause-effect explanation would be an extrapolation of our study. Nevertheless, we expanded the information in our new version.

One minor problem:

Reference numbers need to be moved to in front of “,” or “.”.

ANSWER: Thank you for your attention. Corrected.

Round 2

Reviewer 2 Report

The authors have properly responded to my comments, and the paper has been improved in the process. No further comments.  

This manuscript is a resubmission of an earlier submission. The following is a list of the peer review reports and author responses from that submission.

Round 1

Reviewer 1 Report

The current submission reports the findings from a cross sectional study that focused on Balance Differences between North and South American Older Adults. Although falling in the elderly is a concern, it is difficult to see how the findings add to the current body of knowledge. A stronger rationale for conducting this study is necessary. The main limitation of the study was lacking of falling information. There are numerous events and/or impairments that could have developed between falling (or not falling) and the time of the study so it is difficult to connect the balance measurement results to being actual falling risk. Furthermore, the method section is poorly described, nothing was mentioned about the strategies and the participants' selection process. Only age, height and weight were reported, more detailed information is needed.

Reviewer 2 Report

The authors present a cross-sectional analysis to try to detect differences in balances in patients with various risks of falling across two continents (spread across three countries). The authors report standard statistical measures and capture balance using the BTracks system with subjects eyes closed. The text flows well.

Major issues:

  1. Sampling across genders is imbalanced (2x44 Male vs 2x151 Female). You have 2 genders, 3 risk categories, 2 regions, so 2x3x2=12 groups. Sample size is insufficient for the Kruskal-Wallis Test, which requires a sample size of at least 20 for each group when you have 10-12 groups. Several of the Male categories are undersampled.

Minor issues:

  1. Please add the n’s to Table 2 in addition to the percent’s.
  2. There are an inappropriate number of self-citations to papers written by Daniel Goble, who appears in 8 references (14% of all references in the paper). Some of these are appropriate, but others are probably unnecessary.